# Viscosity-dependent control of protein synthesis and degradation

Yuping Chen [1,3] ✉, Jo-Hsi Huang[1,3], Connie Phong[1] & James E. Ferrell Jr. [1,2] ✉

It has been proposed that the concentration of proteins in the cytoplasm maximizes the speed of important biochemical reactions. Here we have used *Xenopus* egg extracts, which can be diluted or concentrated to yield a range of cytoplasmic protein concentrations, to test the effect of cytoplasmic concentration on mRNA translation and protein degradation. We find that protein synthesis rates are maximal in ~1x cytoplasm, whereas protein degradation continues to rise to a higher optimal concentration of ~1.8x. We show that this difference in optima can be attributed to a greater sensitivity of translation to cytoplasmic viscosity. The different concentration optima could produce a negative feedback homeostatic system, where increasing the cytoplasmic protein concentration above the 1x physiological level increases the viscosity of the cytoplasm, which selectively inhibits translation and drives the system back toward the 1x set point.

The cytoplasm is crowded with macromolecules, with proteins being the most abundant class. The cytoplasmic protein concentration ranges from ~75 mg/mL in mammalian cell lines[1] to 200–320 mg/mL in *E. coli*[2,3]. For a given cell type, the concentration of macromolecules in the cytoplasm is tightly regulated and nearly constant[1,4–6]. This brings up two basic questions: why is the cytoplasmic protein concentration as high as it is, and no higher, and what mechanisms set and maintain this concentration? It has been conjectured that the normal cellular protein concentration maximizes the rates of important biochemical reactions, with there being a trade-off between the effects of concentration on enzyme-substrate proximity and viscosity[7]. Here we set out to directly test the conjecture experimentally.

We chose to use the *Xenopus* egg extract system, an undiluted cell-free living cytoplasm, for these studies because they allow easy, direct manipulation of the concentration of cytoplasmic macromolecules. These extracts carry out the complex biological functions of an intact *Xenopus* egg or embryo faithfully, including self-organization[8–10], DNA replication[11–13], and mitosis[8,14–16]. The processes we chose to examine were protein synthesis and degradation (Fig. 1a), which are not only potentially affected by the cytoplasmic protein concentration, but also directly involved in determining the protein concentration. *Xenopus* extracts can carry out protein synthesis from their own stores of mRNAs or from added synthetic mRNAs[17–19], and they degrade both endogenous proteins (most notably the various substrates of the APC/C)[20,21] and added probe proteins [the present work]. The protein concentration of a *Xenopus* extract is similar to that of mammalian cell lines[22], and the median protein half-lives in *Xenopus* extracts[23] and mouse NIH3T3 cells[24] are almost identical. Moreover, the simplicity, manipulability, and verisimilitude of the extract system makes it an attractive choice for the present studies of how protein synthesis and degradation are affected by the cytoplasmic concentration.

## Results

### *Xenopus* egg extracts are robust towards cytoplasmic dilution and concentration

The most commonly used types of extract are CSF (cytostatic factor)-arrested extracts, interphase-arrested extracts, and cycling extracts[8,22,25,26]. In pilot experiments we found that cycling extracts were the most reliable and longest-lived, and so they were chosen for most of the experiments that follow. One caveat was that previous work has shown that translation rates vary between interphase and M-phase[18], raising the concern that the time course of protein synthesis would be characterized by alternations between different rates. This proved not to be the case, possibly because mitosis sweeps through

[1]Department of Chemical and Systems Biology, Stanford University School of Medicine, Stanford, CA 94305, USA. [2]Department of Biochemistry, Stanford University School of Medicine, Stanford, CA 94305, USA. [3]These authors contributed equally: Yuping Chen, Jo-Hsi Huang. ✉e-mail: cyp7cn@gmail.com; James.ferrell@stanford.edu

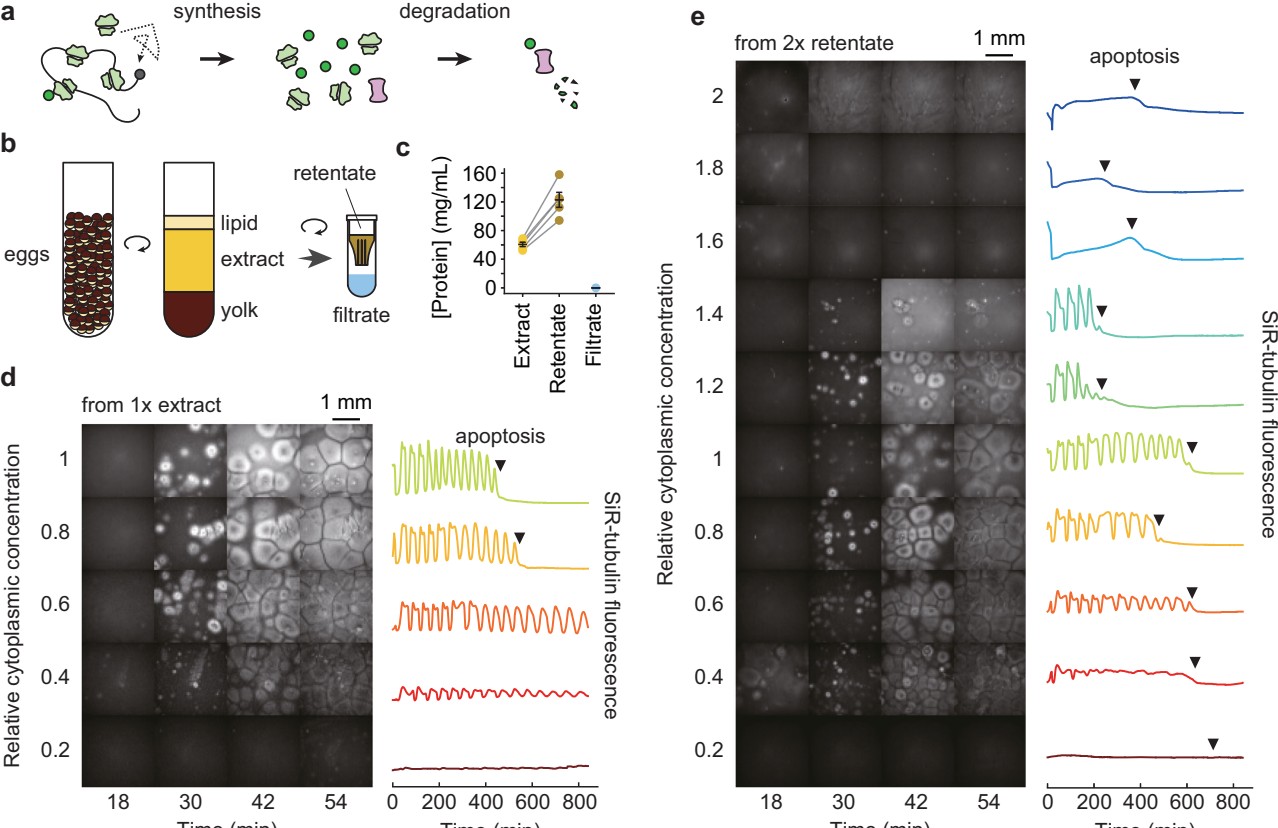

**Fig. 1 | General properties of diluted and concentrated *Xenopus* egg extracts: effects on self-organization and cycling. a** Schematic view of protein synthesis and degradation. **b** Preparation of *Xenopus* egg extract, 2× concentrated retentate, and protein-depleted filtrate. **c** Protein concentration in extract, retentate, and filtrate. Concentrations were determined by Bradford assays. Data are from five extracts. Individual data points are overlaid with the means and standard errors. Source data are provided as a Source Data file. **d** SiR-tubulin staining (left) and SiR-tubulin fluorescence intensity as a function of time (right) in an extract after various dilutions. The starting material was a 1× extract, diluted with various proportions of filtrate, and imaged in a 96-well plate under mineral oil. All fields are shown at equal exposure. The fluorescence intensities shown on the right were quantified from the center 1/9 of the wells. **e** SiR-tubulin staining (left) and SiR-tubulin fluorescence intensity as a function of time (right) in an extract after various dilutions. The data were collected as in (**d**) except that the starting material was a 2× extract.

the extract in spatial waves[27] (see Movies S1 and S2), so that both mitotic and interphase rates probably contribute to the measured rate of synthesis at all time points.

The protein concentration in the cycling extracts used here was found to be $60.7 \pm 3.0$ mg/ml (S.E., $n = 5$), similar to previous measurements of extracts prepared by similar methods (~80 mg/mL[28,29]; ~58 mg/mL[30]). Using 10 kDa Texas Red-dextran as an extracellular marker, from two independent extract preparations we estimated that the final extract was diluted by between 0.4% and 4% by residual extracellular buffer in the packed eggs. The measured extract protein concentration was also close to an often-cited early estimate of the yolk-free cytoplasmic protein concentration in *Xenopus* eggs (~50 mg/mL[31]). Thus the assumption that extracts represent minimally diluted cytoplasm appears to be correct, and we refer to the concentration of these extracts as 1×.

To alter the macromolecular concentration of cycling extracts, we used a spin-column with a 10 kilodalton (kDa) cutoff to produce a cytosolic filtrate and a concentrated cytoplasmic retentate (Fig. 1b). The filtrate was found to be essentially protein-free by Bradford assays as well as by gel electrophoresis and tri-halo compound or Coomassie staining (Fig. 1c and Supplementary Fig. 1). The retentate was on average 2-fold concentrated compared to the starting cytoplasm by Bradford assay (Fig. 1c). We then diluted either the starting 1× extract or the 2× retentate with the filtrate to generate extracts with a range of cytoplasmic macromolecule concentrations (Fig. 1d, e).

We first examined the effects of concentration on two basic aspects of the extract's function: its ability to self-organize and to cycle. The extract's gross organization was found to be robust to cytoplasmic dilution (Fig. 1d, e). Using a microtubule stain, SiR-tubulin, cell-like compartments[8,16] were found to form even when the extract was diluted to as low as 0.3× (Fig. 1d, e, and Supplementary Movies 1–3), in general agreement with a previous report[8], and to partially organize, with small asters, even at 0.2× the normal cytoplasmic concentration (Supplementary Movie 3). By following microtubule polymerization and depolymerization, cell cycles were detected down to dilutions of 0.2×, and the oscillations often persisted for at least 14 h with more than 10 complete periods (Fig. 1c, d, Supplementary Movies 1–3). In agreement with a previous study[32], the diluted extracts did not show signs of significant cell cycle defects except that the duration of interphase and the cell cycle period increased with increasing dilution.

On the other hand, the 2× retentate behaved less normally. The extract was noticeably stickier and more viscous than a 1× extract. Cell-like compartments failed to form at concentrations higher than 1.4× (Fig. 1e), and the cell cycle appeared to be arrested above 1.4×. Nevertheless, when a 2× extract was diluted to 1.4× or less, cell-like compartment formation was restored, and the cell cycle behavior was almost normal (Fig. 1e), although dilutions from the 2× retentate had slightly longer interphases and were slightly more susceptible to apoptosis than dilutions from a 1× extract (Figs. 1d, e). Overall, extracts were more sensitive to being concentrated than diluted, but even so,

extracts were able to carry out self-organization and cycling over a wide range of cytoplasmic concentrations.

## Protein synthesis peaks at a 1× cytoplasmic concentration

To measure the protein synthesis rate, we added an mRNA for eGFP and monitored fluorescence as a function of time. In pilot experiments, we titrated the mRNA concentration and found that we could obtain a satisfactory signal without saturating the translation machinery using a concentration of 2.5 μg/mL (Fig. 2a). We then recorded time courses of eGFP fluorescence intensity for diluted and concentrated extracts all

with the same 2.5 μg/mL concentration of mRNA for eGFP, and calculated translation rates from the linear portion of the time course (Fig. 2b). Figure 2c summarizes the data as directly obtained, with equal concentrations of mRNA at each dilution but differing concentrations of the translation machinery. Translation increased with cytoplasmic concentration to a maximum at ~0.75× and fell thereafter. To relate this to endogenous translation, given that the endogenous mRNAs would vary with concentration just as the translation machinery does, we calculated an inferred endogenous translation rate, taken as the observed translation rate multiplied by the cytoplasmic

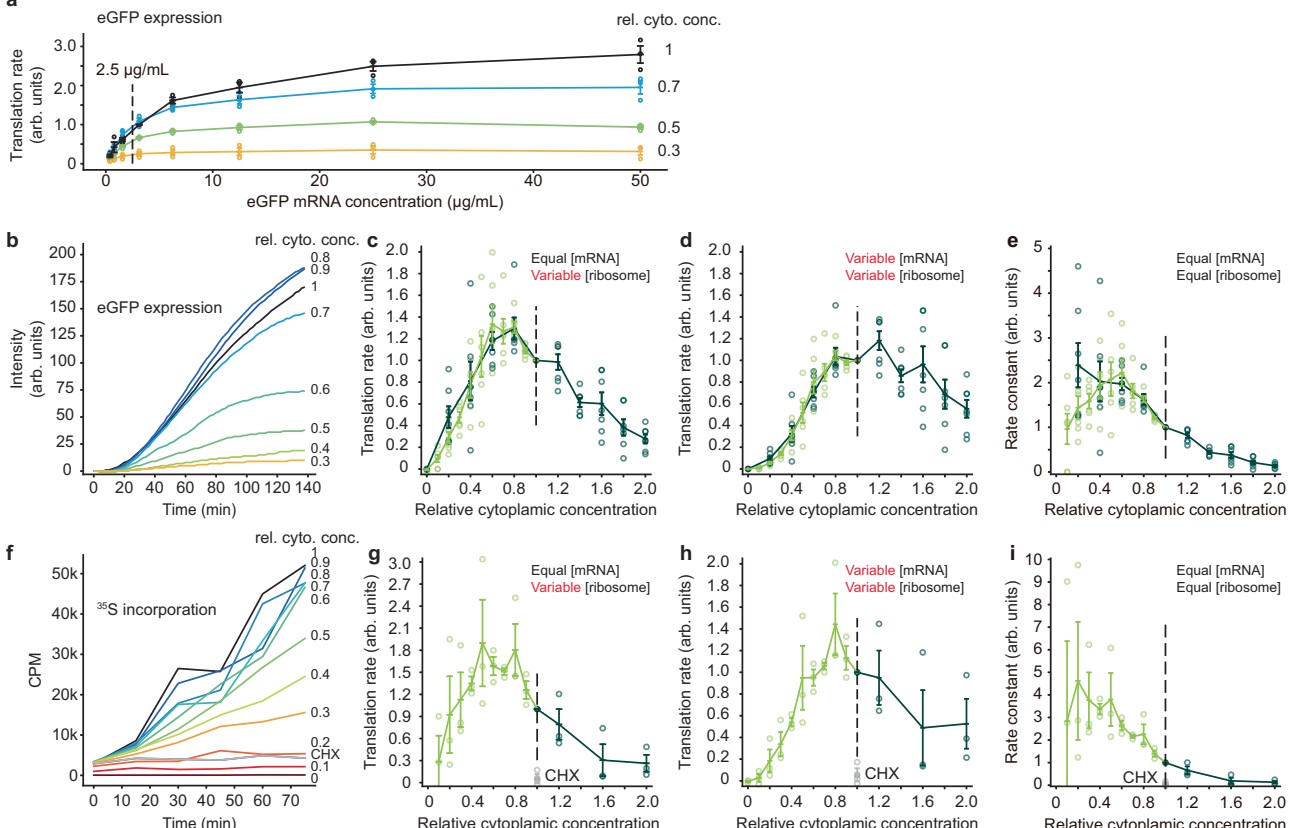

**Fig. 2 | The rate of mRNA translation is maximal at a cytoplasmic concentration of-1x. a** Titration of mRNA concentration for eGFP expression. The indicated concentration (2.5 μg/mL) was chosen for the experiments in (**b**–**e**). Data from $n = 3$ independent experiments. Data are presented as mean values ± SEM. **b** eGFP expression as a function of time for various dilutions of a 1× extract. **c** Translation rate as a function of cytoplasmic concentration. These are the directly-measured data from experiments where the eGFP mRNA concentration was kept constant and the translation machinery was proportional to the cytoplasmic concentration. Data are from $n = 6$ independent experiments for dilution from 1× extracts and $n = 7$ independent experiments for dilution from 2× retentates. Data are normalized relative to the translation rates at a cytoplasmic concentration of 1×. Means and standard errors are overlaid on the individual data points. In this and the subsequent panels, the darker green represents data from diluting 2× retentates and the lighter green from diluting 1× extract. **d** Inferred translation rates for the situation where the mRNA concentration as well as the ribosome concentration is proportional to the cytoplasmic concentration. The rates from (**c**) were multiplied by the relative cytoplasmic concentrations. Data are from $n = 6$ independent experiments for dilution from 1× extracts and $n = 7$ independent experiments for dilution from 2× retentates. Data are presented as mean values ± SEM. **e** Inferred translation rates for the situation where both the mRNA concentration and the ribosome concentration are kept constant at all dilutions. This represents an estimate of the apparent bimolecular rate constant for translation. The rates from (**c**) were divided by the relative cytoplasmic concentrations. Data are from $n = 6$ independent experiments for dilution from 1× extracts and $n = 7$

independent experiments for dilution from 2× retentates. Data are presented as mean values ± SEM. **f** TCA-precipitable $^{35}$S incorporation as a function of time for translation from endogenous mRNAs. Various dilutions of a 1× extract are shown. CHX denotes a 1× extract treated with 100 μg/mL cycloheximide. **g** Inferred translation rates for the situation where mRNA concentration is kept constant and ribosome concentration is proportional to the cytoplasmic concentration. The rates from (h) were divided by the relative cytoplasmic concentration. The gray data points are from CHX (100 μg/mL)-treated 1× extracts. Data are from $n = 3$ independent experiments for dilution from 1× extracts and $n = 3$ independent experiments for dilution from 2× retentates. Data are presented as mean values ± SEM. **h** Translation rate as a function of cytoplasmic concentration. These are the directly measured data from experiments where the $^{35}$S concentration was kept constant but both the (endogenous) mRNA concentration and translational machinery were proportional to the cytoplasmic concentration. Data are from $n = 3$ independent experiments for dilution from 1× extracts and $n = 3$ independent experiments for dilution from 2× retentates. Data are normalized relative to the translation rates at a cytoplasmic concentration of 1×. Means and standard errors are overlaid on the individual data points. **i** Inferred translation rates for the situation where both the mRNA concentration and the ribosome concentration are kept constant at all dilutions. The rates from (**h**) were divided twice by the relative cytoplasmic concentrations (i.e., by the relative concentration squared). Data are from $n = 3$ independent experiments for dilution from 1× extracts and $n = 3$ independent experiments for dilution from 2× retentates. Data are presented as mean values ± SEM. Source data for panels (**a**, **d**, and **g**) are provided as a Source Data file.

concentration. This is shown in Fig. 2d; maximal translation was obtained at a cytoplasmic concentration of -1×. We also calculated the translation rate normalized for both equal ribosome concentration and equal mRNA concentration, by dividing the raw translation rates by the cytoplasmic concentration (Fig. 2e). This provides an estimate of how the apparent bimolecular rate constant of the translation machinery is affected by cytoplasmic concentration. The inferred apparent rate constant fell markedly with increasing cytoplasmic concentration above 0.5×; below that there was too much variability to draw conclusions. These findings show that protein synthesis is fastest in 1× cytoplasm, as predicted by the maximal speed conjecture[7], and suggest that protein synthesis is inhibited when the cytoplasmic viscosity is higher than normal. This latter point is explored further below.

As a second way of gauging the translation rate, we added equal concentrations of $^{35}$S-methionine to extracts with a range of cytoplasmic concentrations and no added exogenous mRNA, and measured $^{35}$S incorporation into TCA-precipitable material. Note that in this experiment, both the mRNA and ribosome concentrations vary with the cytoplasmic concentration. Figure 2f shows that incorporation increased linearly with time through 75 min, and that the protein synthesis inhibitor cycloheximide blocked this incorporation. Translation peaked at a cytoplasmic concentration of -0.8× (Fig. 2h shows the directly observed translation rate; Fig. 2g, i show the inferred rates for constant [mRNA] and constant [mRNA] plus constant [ribosome], respectively). Overall the dependence of translation on cytoplasmic concentration was very similar to that seen with eGFP (compare Fig. 2c and g, 2d and h, and 2e and i), and again the inferred reaction rate constant for translation fell with cytoplasmic concentrations above -0.5x (Fig. 2i). Thus the effects of cytoplasmic concentration on eGFP translation appear to be seen with translation from endogenous mRNAs as well.

## Protein degradation peaks at a higher cytoplasmic concentration

As a first measure of protein degradation, we made use of an exogenous protein substrate, a heavily BODIPY (boron-dipyrromethene)-labeled BSA, DQ-BSA (for dye-quenched bovine serum albumin). This protein becomes fluorescent during degradation because the BODIPY groups become dequenched. We carried out titration experiments, which showed that an approximately linear response could be obtained with a DQ-BSA concentration of 5 μg/mL (Fig. 3a, b). This concentration was then used for experiments with extracts diluted to various extents. As shown in Fig. 3c, the rate of DQ-BSA dequenching increased with the concentration of macromolecules and peaked at about 1.6×. The proteasome inhibitor MG132 blocked this dequenching, indicating that dequenching was largely due to proteasomes rather than lysosomes. As we did for eGFP synthesis, we also multiplied the rate data by the extract concentration to infer a degradation rate for endogenous proteins, where both the substrate and the proteolysis machinery would be affected by cytoplasmic concentration; this shifted the activity peak to -1.8× (Fig. 3d). Note that by paired t-test, the average for the 1.8× data was not significantly higher than the average for the 2× data (p = 0.26 for a one-tailed t-test), so the optimal cytoplasmic concentration for DQ-BSA dequenching may actually be higher than 1.8×. The apparent bimolecular rate constant calculation (Fig. 3e) showed that the enzyme activity fell above -1.4× cytoplasmic concentration. At lower cytoplasmic concentrations, the relationship between this gauge of activity and concentration was complicated; perhaps simple bimolecular kinetics do not pertain in this regime.

As a second measure of protein degradation, we used a securin-CFP fusion protein as a reporter. Securin is a cell cycle protein known to be targeted for proteasome-mediated protein degradation by the anaphase-promoting complex/cyclosome (APC/C) in late mitosis. We measured the decay of securin-CFP fluorescent intensity and calculated the degradation rate[21] for each of the dilution conditions (Fig. 3f).

As was the case with DQ-BSA dequenching, the protein degradation rate peaked at a -1.6× cytoplasmic macromolecular concentration (Fig. 3g), and the inferred rate for a substrate being diluted along with the degradation machinery peaked at 1.8× (Fig. 3h). Again, by paired t-test, the average for the 1.8× data was not significantly higher than the average for the 2× data (p = 0.34 for a one-tailed t-test), so the optimal cytoplasmic concentration for securin-CFP degradation may be higher than 1.8×. The inferred apparent rate constant for securin-CFP degradation fell steadily with concentration (Fig. 3i). Thus, by both measures, protein degradation rates were maximal at a cytoplasmic concentration of about 1.8×, higher than the optimal concentration for protein synthesis.

## Viscosity affects protein translation rate and, to a lesser extent, protein degradation

The decrease in translation at high cytoplasmic macromolecule concentrations suggests that translation is diffusion-controlled in a viscous cytoplasm and that concentrated cytoplasm has a higher viscosity that reduces the molecular movement necessary for the translation reaction. To test these ideas, we first measured how diffusion coefficients vary with cytoplasmic concentration. We used single particle tracking of fluorescently labeled PEGylated 100 nm diameter polystyrene beads, which are larger than proteasomes and ribosomes, and comparable in size to some of the large complexes involved in translation (Fig. 4a). In 1× extracts, the motion of the beads was sub-diffusive, with the diffusivity exponent α equal to $0.88 \pm 0.06$ (means ± S.E.) (Fig. 4b, c). We calculated an average effective diffusion coefficient D from a fit of the random walk diffusion equation to the data over a time scale of 1 s. This was found to be $0.36 \pm 0.23$ μm²/s (means ± S.E.), in good agreement with previous studies[6,33]. There was substantial variability from position to position in the speed of diffusion (Fig. 4b, insets, and c; Supplementary Fig. 3). Similarly high variability has been reported for the diffusion of genetically-encoded nanoparticles expressed in *S. pombe*[34], which is thought to reflect the heterogeneity of the cytoplasmic environment.

The effective diffusion coefficient for the 100 nm beads was highly sensitive to changes in the cytoplasmic concentration (Fig. 4d); D was -11× higher in filtrate (0× cytoplasm) compared to 1× extracts, and -11× lower in 2× extracts. The measured diffusion coefficients obeyed Phillies's law[35,36]:

$$D[\phi] = D[0]e^{-\mu\phi}, \qquad (1)$$

where $D[\phi]$ is the diffusion coefficient at a relative cytoplasmic concentration $\phi$, $D[0]$ is the diffusion coefficient in filtrate, and $\mu$ is a scaling factor that depends upon the size of the probe. A similar relationship between diffusion coefficients and macromolecular concentration has been observed for large multimeric protein complexes[37]. Similarly, the scaling factor varied approximately linearly with the measured size of the beads (Fig. 4e, f)[37,38]. Thus, diffusion was markedly affected by changing the cytoplasmic macromolecule concentration over a 0× to 2× range, and the changes were greatest for large probe particles.

Next, we diluted extracts to 0.7×, 0.8×, or 0.9×, and altered the cytoplasmic viscosity by adding Ficoll 70, a protein-sized (70 kDa) carbohydrate that can act both as a crowding agent and a viscogen (Fig. 4g). The effective diffusion coefficients were found to decrease with increasing Ficoll concentration; 6% Ficoll 70 decreased the diffusion coefficient for a 40 nm bead by a factor of 33, and 2–3% Ficoll 70 yielded diffusion coefficients comparable to those measured in 2× cytoplasm. Thus, Ficoll can be used to bring about the marked changes in viscosity seen when cytoplasm is concentrated, without changing the gross protein concentration.

We therefore asked whether this range of Ficoll 70 concentrations would inhibit protein synthesis and degradation. Note

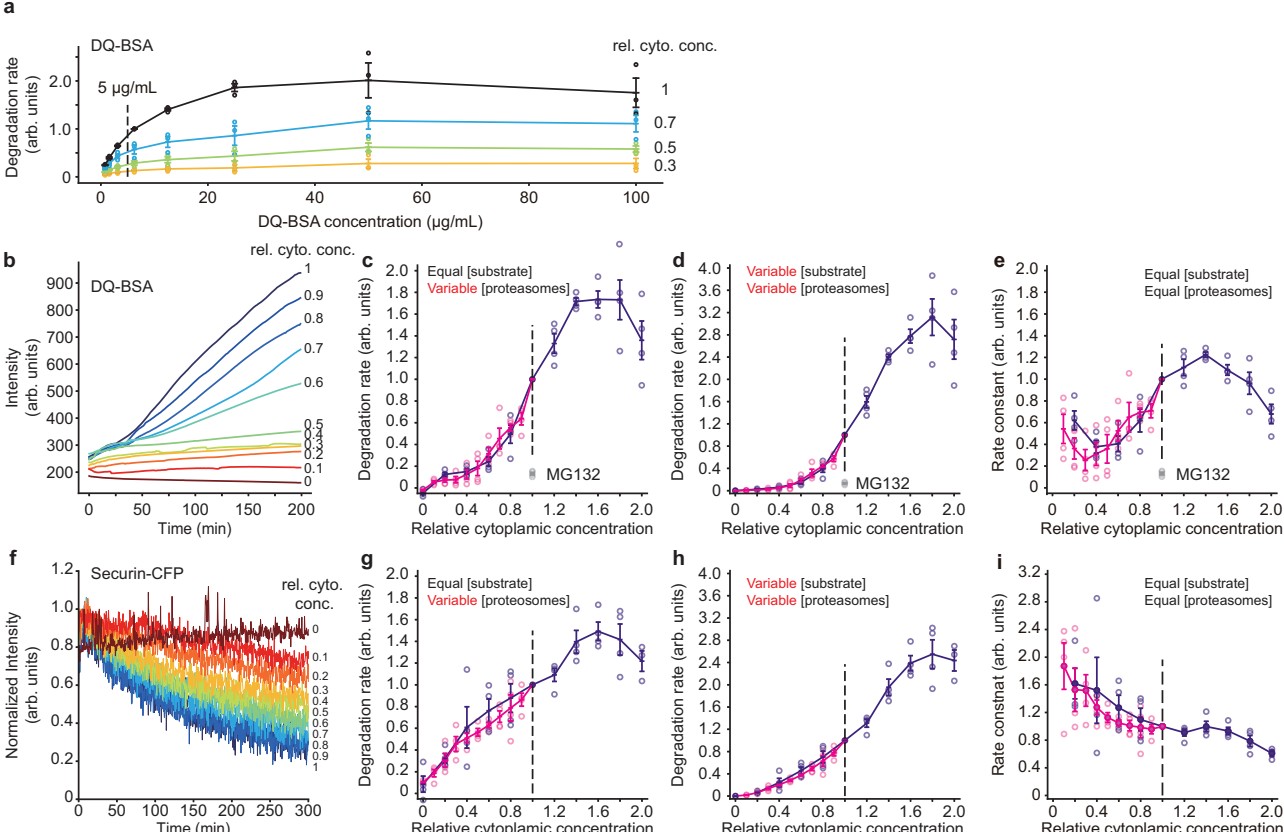

**Fig. 3 | The rate of protein degradation is maximal at cytoplasmic concentrations of~1.8×. a** Titration of substrate protein concentration for DQ-BSA degradation experiments. The indicated concentration (5 µg/mL) was chosen for the experiments in (**b**–**e**). Data from $n = 3$ independent experiments. Data are presented as mean values ± SEM. **b** DQ-BSA fluorescence as a function of time for various dilutions of a 1× extract. **c** Degradation rate as a function of cytoplasmic concentration. These are the directly-measured data from experiments where the DQ-BSA concentration was kept constant and the proteolysis machinery was proportional to the cytoplasmic concentration. The gray data points denoted MG132 are from 1× extracts treated with 200 µM MG132, a proteasome inhibitor. Data are from 4 experiments for dilution from 1× extracts and 4 experiments for dilution from 2× retentates. Data are normalized relative to the degradation rates at a cytoplasmic concentration of 1×. In this and the subsequent panels, the darker purple represents data from diluting 2× retentates and the lighter purple from diluting 1× extract. **d** Inferred degradation rates for the situation where the substrate concentration as well as the proteasome concentration is proportional to the cytoplasmic concentration. The rates from (**c**) were multiplied by the relative cytoplasmic concentrations. Data are from $n = 4$ independent experiments for dilution from 1× extracts and $n = 4$ independent experiments for dilution from 2× retentates. Data are presented as mean values ± SEM. **e** Inferred degradation rates for the situation where both the substrate concentration and the proteasome concentration are kept constant at all dilutions. The rates from (**c**) were divided by

the relative cytoplasmic concentrations. This represents an estimate of the apparent bimolecular rate constant for degradation. Data are from $n = 4$ independent experiments for dilution from 1× extracts and $n = 4$ independent experiments for dilution from 2× retentates. Data are presented as mean values ± SEM. **f** Degradation of securin-CFP as a function of time for various dilutions of a 1x extract. **g** Degradation rate as a function of cytoplasmic concentration. These are the directly-measured data from experiments where the securin-CFP concentration was kept constant but the proteasome concentration was proportional to the cytoplasmic concentration. Data are from 4 experiments for dilution from 1× extracts and 4 experiments for dilution from 2× retentates. Data are normalized relative to the degradation rates at a cytoplasmic concentration of 1×. Means and standard errors are overlaid on the individual data points. **h** Inferred degradation rate for the situation where both the substrate and proteasome concentrations are proportional to the cytoplasmic concentration. The rates from (**g**) were multiplied by the relative cytoplasmic concentrations. Data are from $n = 4$ independent experiments for dilution from 1× extracts and $n = 4$ independent experiments for dilution from 2× retentates. Data are presented as mean values ± SEM. **i** Inferred degradation rates for the situation where both the substrate and the proteasome concentration are kept constant at all dilutions. The rates from (**g**) were divided by the relative cytoplasmic concentrations. Data are from $n = 4$ independent experiments for dilution from 1× extracts and $n = 4$ independent experiments for dilution from 2× retentates. Data are presented as mean values ± SEM. Source data for panels (**a**, **d**, and **h**) are provided as a Source Data file.

that Ficoll might be expected to have either of two opposite effects on enzyme reaction rates: by acting as a crowding agent, it increases the effective concentrations of the reactants and thus could increase the rate of a reaction; but by acting as a viscogen, it could slow protein motions and decrease in the reaction rate. In vitro studies have shown that either of these effects can predominate[39–41]. We found that the rate of translation of eGFP monotonically decreased with increasing Ficoll 70 concentration, with an IC50 of ~2–3% (Fig. 4h). Diffusion coefficients in these Ficoll-supplemented 0.7× to 0.9× extracts supplemented with 2–3% Ficoll 70 were similar to those seen in 2× extracts with no

Ficoll. Thus protein synthesis is sensitive to viscosity over a range relevant to the cytoplasmic concentration/dilution experiments.

The rate of DQ-BSA unquenching was substantially less sensitive to viscosity (Fig. 4i), with an IC50 greater than 6% Ficoll 70. This difference in sensitivity is sufficient to account for the different optimal cytoplasmic concentrations found for translation and degradation.

**A model for the effect of diffusion on reaction rates**
Finally, we asked whether we could derive a simple model to account for the observed rates of protein translation and degradation as a function of cytosolic protein concentration. Assuming mass action

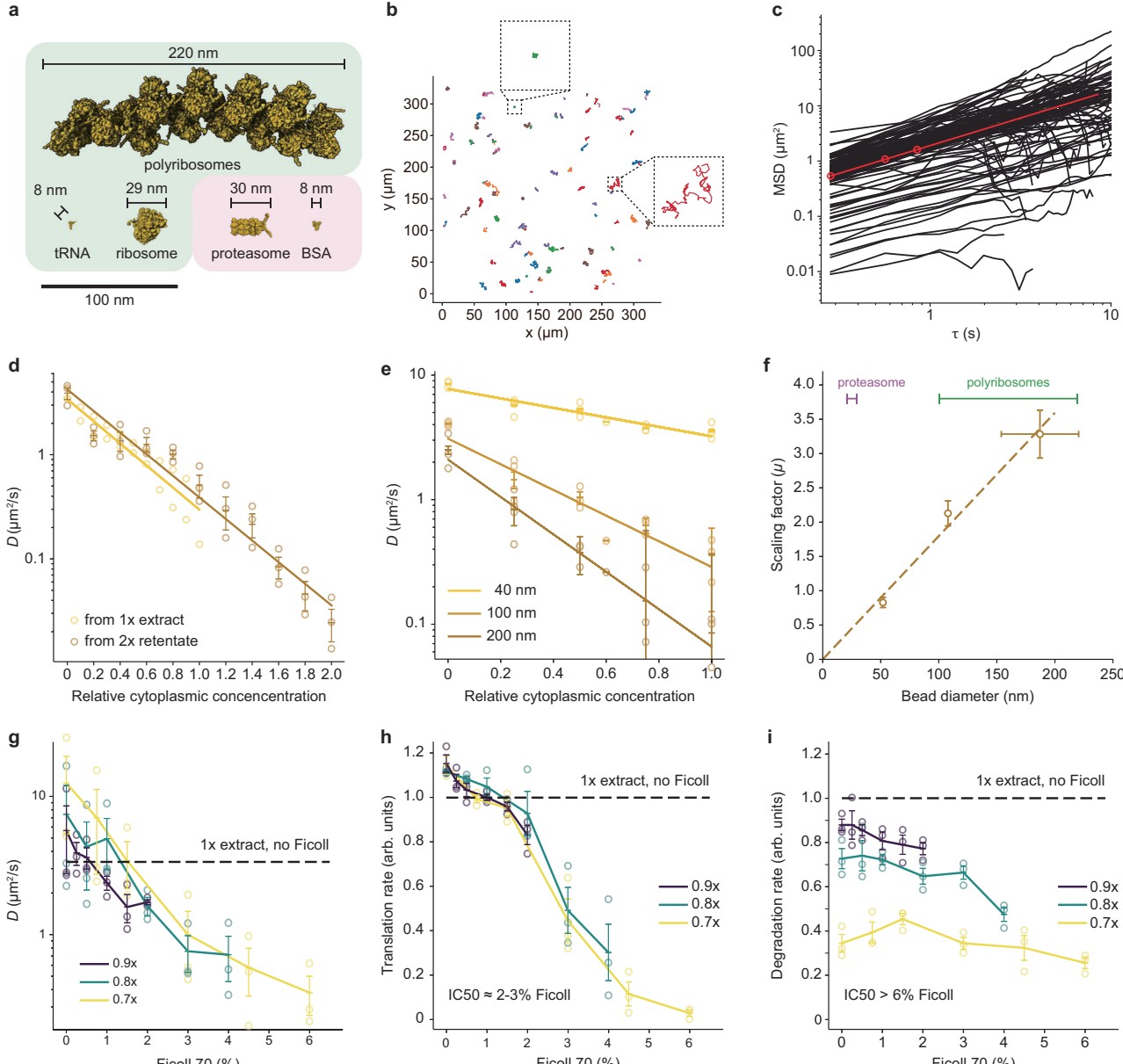

**Fig. 4 | The effect of cytoplasmic concentration on diffusion, and the effect of Ficoll 70 on translation and protein degradation. a** The sizes of various macromolecules and complexes involved in translation and degradation. **b** Single particle traces for diffusion of 100 nm fluorescent beads in 1× cytoplasmic extracts. Two examples of location-to-location variability are highlighted. **c** Mean squared displacement for 110 individual trajectories (black) and average mean squared displacement (red) as a function of the time difference τ. Effective diffusion coefficients were calculated from the first 1 s of data. **d** Effective diffusion coefficients for 100 nm fluorescent beads as function of relative cytoplasmic concentration. Data are from 3 experiments for the 2× extract dilution and from 2 experiments for the 1× extract dilution. Error bars for the 2× extract dilution represent means ± standards errors. **e** Effective diffusion coefficients for beads of different diameter (nominally 40 nm, 100 nm, and 200 nm) as a function of relative cytoplasmic concentration. Data are from 3 experiments. Means and standard errors are overlaid on the individual data points. **f** The scaling factor μ (from Eq. 1) as a function of bead diameter. The apparent bead diameters (nominally 40, 100, and 200 nm) were calculated from their diffusion coefficients in extract buffer with no sucrose using the Stokes-Einstein relationship. Scaling factors are from 3 experiments and are shown as means ± S.E. Bead diameters are from 3 experiments for the 40 nm beads and 4 experiments for the 100 and 200 nm beads, and again are plotted as

means ± S.E. The diameters of proteasomes and polyribosomes are shown for comparison. **g** Diffusion coefficients of 40 nm beads as a function of Ficoll 70 concentration. Extracts were prepared at 0.7×, 0.8×, and 0.9× as indicated and supplemented with Ficoll to yield the final concentrations (w/vol) shown on the x-axis. Data are from 3 experiments. Means and standard errors are overlaid on the individual data points. Diffusion coefficients for the undiluted 1× extracts were also measured and the average is shown for reference. **h** Translation rates, using the eGFP assay, as a function of Ficoll 70 concentration. Extracts were prepared at 0.7×, 0.8×, and 0.9× as indicated and supplemented with Ficoll to yield the final concentrations (w/vol) shown on the x-axis. Data are from the same 3 experiments shown in (**g**). Means and standard errors are overlaid on the individual data points. Translation rates for the undiluted 1× extracts were also measured and the average is shown for reference. **i** Degradation rates, using the DQ-BSA assay, as a function of Ficoll 70 concentration. Extracts were prepared at 0.7×, 0.8×, and 0.9× as indicated and supplemented with Ficoll to yield the final concentrations (w/vol) shown on the x-axis. Data are from the same 3 experiments shown in (**g**). Means and standard errors are overlaid on the individual data points. Degradation rates for the undiluted 1× extracts were also measured and the average is shown for reference. Source data for panels (**d–i**) are provided as a Source Data file.

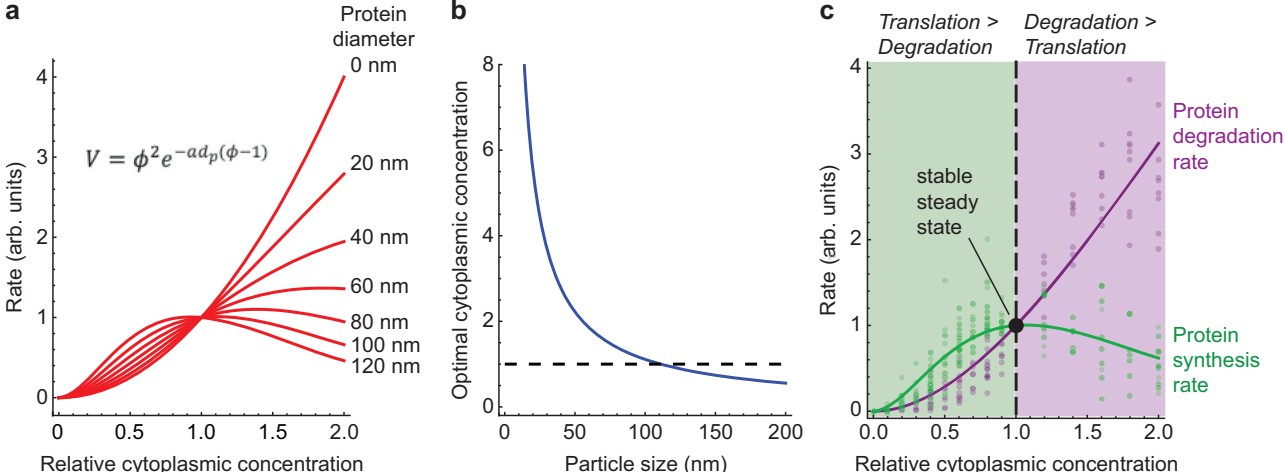

**Fig. 5 | Homeostasis in a model of the effect of cytoplasmic concentration of translation and protein degradation. a** Plot of Eq. 2, which relates a bimolecular reaction rate to the relative cytoplasmic concentration, for various sizes of proteins. We assumed $a = 0.018$ nm$^{-1}$ (from Fig. 4f). **b** Calculated optimal relative cytoplasmic concentration for proteins of different assumed sizes, again assuming $a = 0.018$ nm$^{-1}$. **c** Fits of Eq. 2 to the experimental data for translation (green) and degradation (purple) as a function of cytoplasmic concentration, calculated assuming that both the substrate and enzyme varied with the cytoplasmic concentration. All of the data from Figs. 2d, h, and 3d, h were included in the fits. The $R^2$ values are 0.92 for the translation data and 0.95 for the degradation data. The fitted values for the size of the proteins involved are $104 \pm 2$ nm (mean $\pm$ S.E.) for translation and $14 \pm 1$ nm (mean $\pm$ S.E) for degradation. The fitted optimal cytoplasmic concentrations are $1.07 \pm 0.02$ for translation and $8.1 \pm 0.8$ for degradation (mean $\pm$ S.E.).

kinetics, it follows (Supplementary Materials) that:

$$V = \phi^2 e^{-ad_p(\phi-1)}, \qquad (2)$$

where $\phi$ is the cytoplasmic concentration, $V$ is the reaction rate relative to its rate at $\phi = 1$, $a$ is a scaling factor, and $d_p$, the macromolecular diameter. Note that we have an experimental estimate for $a$ (which, from Fig. 4f is 0.018 nm$^{-1}$), leaving only one adjustable parameter, $d_p$, the macromolecular diameter. Equation 2 defines a biphasic, non-monotonic curve (Fig. 5a), and the larger the assumed macromolecular diameter, the further to the left the curve's maximum lies (Fig. 5, b). The experimentally observed rates for translation and degradation are well captured by Eq. 2, with the fitted values of $d_p$ being $104 \pm 2$ nm for translation and $14 \pm 1$ nm for degradation (means $\pm$ S.E.) (Fig. 5c). Alternatively, we can derive an expression for velocity vs. cytoplasmic concentration for a Michaelis–Menten system rather than a mass action system. This allows us to examine two other factors—the degree to which the system is reaction-controlled versus diffusion-controlled and the extent of enzyme saturation—that can bear upon the position of the optimal cytoplasmic concentration (Supplementary Eq. S20, Supplementary Fig. 4). Like the simpler mass action expression (Eq. 2), it can account for the observed experimental data.

Assuming that the system is in steady state—the translation and degradation rates will be equal—at a relative cytoplasmic concentration of 1×, the steady state is guaranteed to be stable. If the system is perturbed such that the cytoplasmic concentration exceeds 1×, then the degradation rate will rise and the translation rate will fall, driving the system back toward the 1× steady state (Fig. 5c). Conversely, if the cytoplasmic concentration falls below 1×, the translation rate will exceed the synthesis rate, again driving the system back toward the physiological set point (Fig. 5c).

## Discussion

Here we have tested the hypothesis that the concentration of macromolecules in the cytoplasm is set to maximize the rates of important biochemical reactions. We found that in cycling *Xenopus* egg extracts, the rate of translation, as measured by the synthesis of eGFP from an exogenous mRNA and the incorporation of $^{35}$S-methionine into endogenous translation products, does peak at a 1× cytoplasmic concentration, consistent with the maximal speed hypothesis (Fig. 2). This finding fits well with previous studies of cost minimization and near-optimal resource allocations in models of *E. coli* protein synthesis[42–44]. However, protein degradation, as measured by DQ-BSA dequenching and CFP-securin degradation peaks at a higher cytoplasmic concentration, ~1.8× or perhaps higher (Fig. 3). The difference in concentration optima can be explained by the greater sensitivity of translation to increases in viscosity (Fig. 4h, i). This in turn may be due to differences in the sizes of the macromolecular complexes involved in the two processes, as the diffusion coefficients for large fluorescent beads are more affected by cytoplasmic concentration than those of smaller beads (Fig. 4e, f) or, alternatively, translation could be running closer to the diffusion limit than degradation (Fig. 3b). Note that although the conventional wisdom is that almost all biochemical reactions are reaction-controlled (i.e., the catalytic rate constant $k_2$ is much smaller than $k_{-1}$) rather than diffusion-controlled ($k_{-1}$ is much smaller than $k_2$)[45,46], both of the processes measured here were inhibited by the crowding agent/viscogen Ficoll 70, with translation being substantially more sensitive than degradation.

The different optimal cytoplasmic concentrations for translation vs. degradation mean that the system is homeostatic: increasing the concentration of macromolecules in the cytoplasm would increase the rate of degradation and decreases the rate of translation, whereas decreasing the cytoplasmic concentration would decrease the rate of degradation (Fig. 5). This behavior can be explained through a theoretical treatment based on mass action kinetics and Phillies's law[35,36].

Note that the shapes of the rate curves mean that protein synthesis and degradation can be viewed as a negative feedback system. Increasing the cytoplasmic protein concentration increases the cytoplasmic viscosity, which negatively affects translation, which makes the protein concentration eventually drop—a negative feedback loop based on the sensitivity of translation to viscosity. Conversely, decreasing the cytoplasmic protein concentration decreases the cytoplasmic viscosity, which mitigates the drop in translation rates that would normal be expected from a decrease in ribosome and mRNA concentration. Such feedback regulation may also pertain to the oscillation of biomass growth rate and maintenance of cytoplasmic density found in single mammalian cells[47]. Negative feedback is a common theme in homeostatic systems. This process can alternatively

be viewed as a variation on end-product inhibition, where the product of protein synthesis inhibits translation, but through the intermediacy of changes in protein diffusion rates rather than through the direct binding of the product to the enzyme.

We do not yet have direct evidence for the time scale of this proposed protein concentration homeostasis. Based on the measurement of 43 h for the median half-life of a *Xenopus* protein during embryogenesis[23], we suspect that the response might require tens of hours. This time scale would be particularly appropriate for protein homeostasis in the immature oocyte, the egg's immediate precursor in development. The oocyte is thought to live for weeks or months in the frog ovary and to vary little in terms of size, appearance, and composition during this time[48,49].

## Methods
### Extract preparation
All *Xenopus* experiments and animal care followed protocols (APLAC-13307) approved by the Institutional Animal Care and Use Committee (IACUC) of Stanford University. *Xenopus laevis* were at least 3 years old from Nasco. Cycling extracts were prepared as described previously[22,50] with the following modifications. Briefly, freshly laid frog eggs were collected, washed with 20 g/L L-cysteine pH 7.8, and incubated for 3–5 min to remove the jelly coat. The eggs were then washed twice with ~150 mL 0.2× MMR solution (20 mM NaCl, 1 mM HEPES, 400 μM KCl, 400 μM CaCl$_2$, 200 μM MgCl$_2$, and 20 μM EDTA pH 7.8) and resuspended in 50 mL 0.2× MMR solution. Calcium ionophore A23187 (C7522, Sigma) was added to a final concentration of 0.5 μg/mL to activate the eggs. After 2 min of activation, liquid was removed, and the eggs were washed twice with ~150 mL 0.2× MMR solution and three times with ~150 mL extract buffer [100 mM KCl, 50 mM sucrose, 10 mM HEPES pH 7.7 (with KOH), 1 mM MgCl$_2$, and 100 μM CaCl$_2$]. Twenty min after activation (>80% of the eggs showed contraction of the animal pole), the eggs were transferred to a 14 mL round-bottom polypropylene tube (352059, Corning) and packed for 1 min at 300 × *g*. Excess liquid on top of the eggs was removed, and the egg-containing tube was chilled on ice. The eggs were crushed by centrifugation at 16,000 × *g* for 15 min at 4 °C. The cytoplasmic layer was then collected by puncturing the side of the extract-containing tube at ~2 mm above the interface between the extract layer and the yolk layer. The extract was allowed to flow into a collecting Eppendorf tube by gravity or by gently pressing the tube opening with one finger to create a positive pressure inside the tube. The collected extract was mixed with 10 μg/mL leupeptin, 10 μg/mL pepstatin, 10 μg/mL chymostatin, and 10 μg/mL cytochalasin B, and further refined by centrifugation at 16,000 × *g* for 5 min at 4 °C using a tabletop refrigerated centrifuge. After the refining centrifugation, the clarified extract was transferred to a new tube.

CSF extracts[21,22] were prepared similarly to cycling extracts, with the differences being that the eggs were washed with ~150 mL CSF extract buffer (100 mM KCl, 50 mM sucrose, 10 mM potassium HEPES pH 7.7, 5 mM EGTA pH 7.7, 2 mM MgCl$_2$, and 0.1 mM CaCl$_2$) four times immediately after dejellied, without 0.2× MMR washes or calcium ionophore activation. The whole process between dejellying and the crushing spin typically took ~10 min. After the refining centrifugation, the extract was transferred to a new tube and supplemented with 100 μg/mL cycloheximide.

### Filtrate and retentate preparation
Extract (400 μL) was transferred to a 10 kDa molecular weight cut-off centrifugal filter unit (UFC501096, Millipore) placed in a collection tube and centrifuged at 16,000 × *g* for 10 min at 4 °C three times. After each centrifugation period, the extract was taken out of the centrifuge and mixed by pipetting. The filtrate was collected in the collection tube. The retentate was collected by inverting the filtration unit in a new collection tube and centrifuged at 16,000 × *g* for 1 min.

### SiR-tubulin intensity
To monitor microtubules in cycling extracts, 200 nM SiR-tubulin was added to the extract, retentate, and filtrate. Then the extract or retentate was mixed with different volume fractions of filtrate to create a series of dilution conditions. Five μL of the dilutions were loaded onto a 96-well polystyrene assay plate (3368, Corning) and gently spread using the pointed end of the loading pipette tip to achieve an even coverage of extract at the bottom of the well. A layer of 100 μL heavy mineral oil (330760, Sigma) was pipetted to cover the extract and prevent evaporation. The 96-well plate was immediately loaded onto an inverted epifluorescence microscope (DMI8, Leica) for imaging at a frame rate of 0.5 min$^{-1}$. The median intensity from the center 1/9 of each frame was obtained with Leica Application Suite X and analyzed with custom code available on GitHub [https://github.com/yupchen/viscosity_paper], and was plotted with offsets in Fig. 1d, e.

### eGFP translation and DQ-BSA degradation
For eGFP translation and DQ-BSA degradation experiments, the extract, filtrate, or retentate was mixed with 2.5 μg/mL (unless otherwise stated) eGFP mRNA (L-7201-100, Trilink Biotechnologies) or 5 μg/mL DQ-BSA (D12050, Thermo Fisher) on ice. The extract or retentate was mixed with different proportions of the filtrate to generate different dilutions. The dilutions (15 μL) were then added to a clear bottom 384-well plate (324021, Southern Labware) and equilibrated to room temperature. The imaging plate was then loaded onto an inverted fluorescence microscope for time course measurements at a frame rate of 1 min$^{-1}$ or 0.5 min$^{-1}$.

To calculate the rate of eGFP protein synthesis and DQ-BSA degradation, the median intensity of the center quarter of each frame was measured. The raw rates were extracted by calculating the slope of a linear segment from the intensity-time plot. The linear segment was typically between 50 and 120 min for eGFP translation, and between 25 and 120 min for DQ-BSA degradation experiments. Intensity trajectories were manually inspected and the linear segments were adjusted individually to ensure linearity. The raw rates from dilutions of 1× extract (or 2× retentate) were normalized by the rate measured in the original 1× extract (or the reconstituted 1× extract) from the same batch of eggs to control for variability due to batch variation of the eggs and experimental conditions, allowing comparison among experiments.

### $^{35}$S-methionine labeling
A cycling extract was concentrated as described above. The extract, retentate, and filtrate were supplemented with 1% v/v $^{35}$S-methionine to a final concentration of ~0.5 μCi/μL. The filtrate was then mixed with different volume fractions of extract or retentate to generate a series of dilutions. For the "CHX" sample, 100 μg/mL cycloheximide was added to a 1× extract. The extract was then sampled at 15-min intervals, and the translation process was halted by directly mixing 5 μL samples with 100 μL H$_2$O, which was then mixed with 100 μL 50% TCA (trichloroacetic acid, T0699, Sigma). To collect and clean up the TCA-precipitable material, 50 μL of the homogeneous extract/TCA mixture was passed through a glass fiber filter (WHA1820025, Sigma) pre-wetted with 5% TCA, then 1 mL of 5% TCA was passed through the filter to remove soluble material, and the filter was washed with 2 mL of 95% ethanol and dried on vacuum. The filter with collected material was dropped into a 20-mL scintillation vial (03-337-2, Thermo Fisher) containing 10 mL scintillation fluid (111195, RPI Research Products). The radioactivity was measured using a liquid scintillation counter. The rate was calculated similarly to the eGFP translation and DQ-BSA degradation experiments.

### Securin-CFP degradation
The securin-CFP degradation experiments followed a previous protocol[21] with modifications. The fluorescent probe, securin-CFP, was

**Article** https://doi.org/10.1038/s41467-024-46447-w

made by mixing 10 µg of an SP6-securin-CFP plasmid in 20 µL H$_2$O with 30 µL SP6 High-Yield Wheat Germ Protein Expression System (TnT® L3261, Promega), and incubating at room temperature for 2 h per the manufacturer's instructions. CSF extracts were used for these experiments. The CSF extract was additionally supplemented with purified recombinant nondegradable Δ90 sea urchin cyclin B protein (with two additional deletion mutations at L53 and Q54 compared to the previous protocol) at a concentration capable of driving the extract into an M-phase arrest and incubated at room temperature for 30 min. 0.8 mM CaCl$_2$ was added to the extract and incubated for an additional 30 min to degrade endogenous cyclin B. The extract was then divided into two fractions, one kept on ice and the other concentrated using the previously stated method. A series of dilutions were reconstituted by mixing the filtrate with either the extract kept on ice or retentate from the concentrator. 19 µL of the extracts were added and mixed with 1 µL of the in vitro transcribed and translated securin-CFP and pipette into a glass-bottomed 384-well plate. As a background control, we also included a well of extract with no securin-CFP. The time courses of fluorescence intensity were recorded using a fluorescence plate reader at a rate of 2 min$^{-1}$.

To calculate the degradation rate, each experimental reading was subtracted by the corresponding background measurement. The first few data points typically increased with time, possibly due to equilibration of the fluorophore. Therefore, instead of normalizing to the first data point in each time series, the background corrected intensities were divided by the maximum of the first 15 time points (7.5 min). The normalized intensities were fitted to $A = A[0] e^{-kt} + C$, where $A$ is the fluorescence, $k$ is the rate constant, and $t$ is time, and $A[0]$ (constrained to be greater than 0.95), $k$, and C (constrained between 0 and 0.05) are fitting variables. The value of $k$ measured for each dilution condition from 1× extract (or from 2× retentate) was normalized to 1× extract (or nominal 1× reconstituted from the 2× retentate) from the same experiment to control for variations among batches of eggs.

### Single particle tracking
PEGylated fluorescent particles were prepared by mixing 50 µL of 20 mg/mL methoxypolyethylene glycol amine 750 (07964, Sigma), 5 µL of fluorescent polystyrene nano beads (2% solid, F8888 and F8795, Thermo Fisher), 50 µL of 30 mM N-hydroxysulfosuccinimide (56485, Sigma) in 200 mM borate buffer pH8.2, and 10 µL of 100 mM N- (3-dimethylaminopropyl)-N'-ethylcarbodiimide hydrochloride (03450, Sigma). The PEGylation reaction was incubated at room temperature for 20 h. To stop the reaction, the mixture was then diluted 100-fold with water, dialyzed against 3 M NaCl, and then water three times. The particles were further diluted to make it so that a 1:100 dilution provided a suitable concentration for segmentation and tracking. 1:100 (v:v) of the beads were mixed into the extract by pipetting. 5 µL of the extract was placed in the center of a well in a glass-bottom 96-well plate. The plate was loaded onto an inverted epifluorescence microscope. The extract was allowed to equilibrate with the environment for 5 min, and a movie was taken at the appropriate wavelength at a frame rate of 3 Hz using a 40× objective.

The time-lapse videos of particle movements were analyzed with a custom Python script. Briefly, the images were flat field corrected (background flat fields were generated using basic, an ImageJ package) and bleach corrected. The particles were called and linked using the Trackpy library with adaptive mode, which allows calling particle movements with large step size variations. Typical starting parameters for Trackpy were: diameter = 15, maxsize = 7, minmass = 650, search_range = 30, ecc_threshold = 1, percentile = 99.5, topn = 300, memory = 1; drift correction was typically off unless the movie had a translational flow. Movies were discarded if they contained a strong convergent or divergent flow. Parameters were adjusted for individual movies to allow capturing the greatest number of tracks without sacrificing tracking quality. The mean squared displacement for an individual trajectory was calculated by $MSD_i(n\Delta t) = \frac{1}{N-n-1}\sum_{i=1}^{N-n-1}[x(i\Delta t+n\Delta t)-x(i\Delta t)]^2+[y(i\Delta t+n\Delta t)-y(i\Delta t)]^2$, where $N$ is the number of frames in a trajectory and $x$ and $y$ are the coordinates at $i\Delta t$ or $i\Delta t + n\Delta t$. The ensemble mean squared displacement was calculated by $MSD(n\Delta t) = \sum_{i=1}^{K}(N_i - n - 1) MSD_i(n\Delta t)$, where $K$ is the number of trajectories, $N_i$ is the number of frames for the $i$th trajectory, and $MSD_i(n\Delta t)$ is the individual MSD of the $i$th trajectory for a time lag of $n\Delta t$.

### Estimation of the effective diffusion coefficient for the time scale of 1s
To calculate the effective diffusion coefficient, $D_{eff}$, a linear fit was made to the first 3 values to the ensemble MSD vs τ plot (corresponding to τ = 1/3, 2/3, and 1 s). The slope of the fitted line was calculated to obtain MSD/τ. The effective diffusion coefficient was calculated by $D_{eff}$ = MSD/ (4 τ).

### Particle size estimation
PEGylated particles were resuspended in an extract buffer without sucrose. The effective diffusion coefficient for each type of particle was measured by particle tracking as above. The diameters of the particles were calculated using a rearrangement of the Stokes-Einstein equation: $d_p = k_B T/ (3\pi\eta D_{eff})$, where $k_B$ is the Boltzmann constant (1.380649·10$^{-23}$ N m K$^{-1}$), $T$ is temperature (296.15 K), and $\eta$ is the viscosity of extract buffer without sucrose (assumed to be similar to water at 0.001 N m$^{-2}$ s).

### Reporting summary
Further information on research design is available in the Nature Portfolio Reporting Summary linked to this article.

## Data availability
The authors declare that the data supporting the findings of this study are available within the paper and its supplementary information files. Source data are provided with this paper.

## Code availability
Code used in this study is available online on Github [https://github.com/yupchen/viscosity_paper] Chen, Y. (2024). Viscosity-dependent control of protein synthesis and degradation (Version 1.0.0)

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

## Acknowledgements

We thank Ken Dill, Julia Kamenz, Joël Lemière, Jan Skotheim and the Skotheim lab, Michael Zhao and the Ferrell lab for comments. This work was supported by a grant from the NIH (R35 GM131792) to J.E.F.

## Author contributions

Conceptualization, Y.C., J.H. and J.E.F.; Methodology, Y.C., J.H, C.P. and J.E.F.; Software, Y.C.; Formal Analysis, Y.C. and J.E.F.; Investigation, Y.C.; Resources, C.P.; Writing—Original Draft, Y.C. and J.E.F.; Visualization, Y.C. and J.E.F.; Supervision, J.E.F.; Funding Acquisition, J.E.F.

## Competing interests

The authors declare no competing interests.
