## [Peer Review File · Nature Communications]

Viscosity-dependent control of protein synthesis and degradationReviewer #1 (Remarks to the Author):

In this study, Chen et al. investigate the impact of protein concentration on the rates of protein synthesis and degradation. Their findings reveal that protein synthesis is most efficient at approximately one-fold (1x) cytoplasmic concentration, while protein degradation peaks at roughly 1.8 times (1.8x) the cytoplasmic concentration. Additionally, the study examines the effect of varying viscosity levels. The authors propose that the observed differences in concentration optimality may be linked to a heightened sensitivity of the translation process to viscosity changes. Overall, this is a well-conducted and insightful study on a significant topic. In my view, the paper is essentially ready for publication as it stands. My main suggestion is to make the abstract clearer with some minor revisions e.g., see outlined below.

Minor points:

"whereas protein degradation continues to rise to an optimal concentration of $\sim 1.8x$ "

Please rephrase to make it more obvious that 1.8x means in this sentence in the abstract. When I read it for the first time, I was not sure if this 1.8 x refers to concentration or dilution of cytoplasm.

"This can be attributed to the greater sensitivity of translation to cytoplasmic viscosity". Please clarify what "This" refers to.

Please make clear that you tested the change of viscosity experimentally rather than being pure inference in this sentence "This can be attributed to the greater sensitivity of translation to cytoplasmic viscosity, perhaps because it involves large macromolecular complexes like polyribosomes." e.g., by adding. "We show that..."

Please make clearer that the last sentence of the abstract at this point is mostly speculation. Because of the involved timescales I doubt that this is physiologically relevant but it's an important and interesting topic to discuss: "The different concentration optima set up a negative feedback homeostatic system, where increasing the cytoplasmic protein concentration above the 1x physiological level increases the viscosity of the cytoplasm, which selectively inhibits translation and drives the system back toward the 1x set point."

Please check the use of grossly in two sentences below. I am not a native speaker, but usage seems off to me: "Grossly, it was stickier and more viscous than a 1x extract."

"extracts were able to carry out grossly normal self-organization."

An implicit assumption throughout the paper seems to be that *Xenopus* egg extract is 1x native concentration. Could you please state this assumption and/or provide evidence for it. Cytoplasm might be somewhat diluted during extract preparation.

Any insight why the labeled BSA is degraded in extract?

Reviewer #2 (Remarks to the Author):

The study by Chen et al. reports on experiments that test the idea that typical cellular concentrations are optimal for essential biochemical processes. This is studied for *Xenopus* egg extracts that are diluted or concentrated to different degrees. By combining measurements of the protein synthesis rate, the protein degradation rate (both for endogenous and exogenous protein) with measurements of diffusion, the authors conclude that the conjectured concentration optimum is indeed seen. In addition, they find a homeostatic mechanism for the overall protein concentration, since the optimal density for synthesis and degradation is different, so that deviations from the balanced conditions are corrected automatically.

This is a well-designed and carefully done study. Several questions that came to my mind while reading it, were answered shortly afterwards. Overall, I am in favor of its publication. Nevertheless, I have a few comments that the authors should address in a revision.

1) How well defined are the concentrations that are obtained from the dilutions from the 2x concentration vs. from the original 1x? My impression is that conditions with nominally the same concentration obtained from 1x and 2x are similar, but not quite the same (from fig. 1d,e). For the rates in Figs. 2 and 3, I am less sure (also is the normalization the same or individually for the two cases?) - but maybe this is because I had trouble distinguishing the two shades of green. Maybe this comparison could be made more quantitative.

2) Related to 1), I really had problems distinguishing the two types of green in fig. 2c-i. I think this could be colored more clearly.

3) I was wondering whether the exogenous protein in the synthesis and degradation measurements changes the viscosity. Or is the additional concentration very small?

4) The claim that the 100nm beads used for the diffusion measurements are similar in size to the relevant complexes is not convincing in my opinion. My understanding is that polyribosomes form on the mRNA, so the limiting diffusion is probably not for polyribosomes. I may be wrong and this is different for *Xenopus*.

In any case, I do like the explicit diffusion measurements, and 100nm is probably as small as you can go easily with beads.

5) The diffusion measurements show considerable variability (fig. 4c), which in itself is also an interesting observation. I would have liked to see histograms of the diffusion coefficient in addition to just average values. Is there any structure in this broad distribution, e.g. bimodality?

6) The assumption that k_{-1} varies with the diffusion coefficient (above eq. S19) could be justified or explained better.

REVIEWER COMMENTS

Reviewer #1 (Remarks to the Author):

In this study, Chen et al. investigate the impact of protein concentration on the rates of protein synthesis and degradation. Their findings reveal that protein synthesis is most efficient at approximately one-fold (1x) cytoplasmic concentration, while protein degradation peaks at roughly 1.8 times (1.8x) the cytoplasmic concentration. Additionally, the study examines the effect of varying viscosity levels. The authors propose that the observed differences in concentration optimality may be linked to a heightened sensitivity of the translation process to viscosity changes. Overall, this is a well-conducted and insightful study on a significant topic. In my view, the paper is essentially ready for publication as it stands. My main suggestion is to make the abstract clearer with some minor revisions e.g., see outlined below.

Minor points:

“whereas protein degradation continues to rise to an optimal concentration of ~1.8x”. Please rephrase to make it more obvious that 1.8x means in this sentence in the abstract. When I read it for the first time, I was not sure if this 1.8 x refers to concentration or dilution of cytoplasm.

We have made corresponding changes in the Abstract (lines 29-30)

“This can be attributed to the greater sensitivity of translation to cytoplasmic viscosity”. Please clarify what “This” refers to.

We have clarified this as suggested (line 30).

Please make clear that you tested the change of viscosity experimentally rather than being pure inference in this sentence “This can be attributed to the greater sensitivity of translation to cytoplasmic viscosity, perhaps because it involves large macromolecular complexes like polyribosomes.” e.g., by adding. “We show that...”

We have clarified this as suggested (line 30). We also deleted the clause “perhaps because it involves large macromolecular complexes like polyribosomes” because it is also possible that the difference is that protein synthesis runs closer to being diffusion limited than protein degradation does.

Please make clearer that the last sentence of the abstract at this point is mostly speculation. Because of the involved timescales I doubt that this is physiologically relevant but it’s an important and interesting topic to discuss: “The different concentration optima set up a negative feedback homeostatic system, where increasing the cytoplasmic protein concentration above the 1x physiological level increases the viscosity of the cytoplasm, which selectively inhibits translation and drives the system back toward the 1x set point.”

As suggested we have changed the wording from “The different concentration optima set up a negative feedback homeostatic system...” to “The different concentration optima could produce a negative feedback homeostatic system....” (lines 32-33).

Please check the use of grossly in two sentences below. I am not a native speaker, but usage seems off to me: “Grossly, it was stickier and more viscous than a 1x extract.” “extracts were able to carry out grossly normal self-organization.”

Interesting point! In medicine especially the term “grossly” is often used to mean “macroscopically”, e.g. “On autopsy the liver was grossly normal”. But in everyday English it typically means something more like “excessively”. We have eliminated both uses of the term “grossly” (lines 110 and 117).

An implicit assumption throughout the paper seems to be that Xenopus egg extract is 1x native concentration. Could you please state this assumption and/or provide evidence for it. Cytoplasm might be somewhat diluted during extract preparation.

We have estimated the dilution that occurs during extract preparation to be between 0.4 and 4%—i.e., quite minimal. We have added this information to the Results section (lines 77-84)

Any insight why the labeled BSA is degraded in extract?

The exact mechanism of DQ-BSA degradation is not entirely understood. However, DQ-BSA fluorescence is likely a result of proteolysis through a proteasome mediated pathway. In the presence of proteasome inhibitor MG135, BODIPY fluorescence intensity does not increase even in 1x cytoplasm where minimal perturbation was introduced as shown in Fig. 3.

The rapid turnover of DQ-BSA makes us suspect that perhaps the heavy BODIPY labeling of DQ-BSA makes it less stable than unlabeled BSA would be. But we have not pursued this idea.

Reviewer #2 (Remarks to the Author):

The study by Chen et al. reports on experiments that test the idea that typical cellular concentrations are optimal for essential biochemical processes. This is studied for Xenopus egg extracts that are diluted or concentrated to different degrees. By combining measurements of the protein synthesis rate, the protein degradation rate (both for endogenous and exogenous protein) with measurements of diffusion, the authors conclude that the conjectured concentration optimum is indeed seen. In addition, they find a homeostatic mechanism for the overall protein concentration, since the optimal density for synthesis and degradation is different, so that deviations from the balanced conditions are corrected automatically.

This is a well-designed and carefully done study. Several questions that came to my mind while reading it, were answered shortly afterwards. Overall, I am in favor of its publication. Nevertheless, I have a few comments that the authors should address in a revision.

1) How well defined are the concentrations that are obtained from the dilutions from the 2x concentration vs. from the original 1x? My impression is that conditions with

nominally the same concentration obtained from 1x and 2x are similar, but not quite the same (from fig. 1d,e). For the rates in Figs. 2 and 3, I am less sure (also is the normalization the same or individually for the two cases?) - but maybe this is because I had trouble distinguishing the two shades of green. Maybe this comparison could be made more quantitative.

A constant doubling in protein concentration was observed in the retentate for proteins as shown in the pairwise comparisons in Fig. 1c. Regardless of whether nominal or absolute concentration was used, we observe similar trends in the normalized rates as shown in supplementary Fig. S2.

The reviewer is correct that the cell cycle periods in Fig 1 are a little longer for the extracts diluted from 2x vs 1x. For a final concentration of 1x, the periods were 33 min vs. 50 min for extracts diluted from 1x and 2x; for a final concentration of 0.8x they were 44 min vs 57 min, and for a final concentration of 0.6x they were 50 min vs 57 min. From the protein synthesis and degradation measurements, it is less clear whether the extracts diluted from 1x and 2x behaved differently, and it is clear that the overall trends were similar regardless of whether 1x or 2x extracts were used as the starting material, and regardless of which synthesis assay or degradation assay was used.

2) Related to 1), I really had problems distinguishing the two types of green in fig. 2c-i. I think this could be colored more clearly.

We have changed the coloring as suggested.

3) I was wondering whether the exogeneous protein in the synthesis and degradation measurements changes the viscosity. Or is the additional concentration very small?

The concentrations of the added proteins are indeed small compared to the total protein concentration. E.g. DQ-BSA was 5 µg/mL final concentration compared to an endogenous protein concentration of ~60 mg/mL.

4) The claim that the 100nm beads used for the diffusion measurements are similar in size to the relevant complexes is not convincing in my opinion. My understanding is that polyribosomes form on the mRNA, so the limiting diffusion is probably not for polyribosomes. I may be wrong and this is different for Xenopus.

In any case, I do like the explicit diffusion measurements, and 100nm is probably as small as you can go easily with beads.

This is a good point. We now explicitly point out in the text that these beads are bigger than proteosomes and ribosomes (line 196).

5) The diffusion measurements show considerable variability (fig. 4c), which in itself is also an interesting observation. I would have liked to see histograms of the diffusion coefficient in addition to just average values. Is there any structure in this broad distribution, e.g. bimodality?

We have added the histograms as a new supplemental Fig S3. DIP unimodality test gave *p*-values of 0.991;0.964;0.998;0.651;0.426;0.030 for dilutions of 1x, 0.8x, 0.6x,

0.4x, 0.2x, and filtrate. Only the 0x filtrate is unlikely to be behaving unimodally by chance; the others may well exhibit some degree of bi/multimodality.

6) The assumption that k_{-1} varies with the diffusion coefficient (above eq. S19) could be justified or explained better.

This assumption is based on one type of model for protein-protein dissociation—that the rate of dissociation is inversely proportional to the time it takes for the proteins to diffuse a distance r apart, where r is taken to be something on the order of a protein radius or diameter. We now spell this out (lines 824-825) and refer to a review article (C. DeLisi, The biophysics of ligand-receptor interactions. Q. Rev. Biophys. 13, 201-230 (1980).) that goes into detail on these issues.

Reviewer #2 (Remarks to the Author):

The remaining minor points have been clarified. I recommend to accept the paper.